# Study on Long-Term Care Service Awareness, Needs, and Usage Intention of Older Adult Male Homosexuals in Taiwan and Their Ideal Long-Term Care Service Model

**DOI:** 10.3390/healthcare12040418

**Published:** 2024-02-06

**Authors:** Hong Hong

**Affiliations:** Bachelor Program of Senior Health Promotion and Care Management for Indigenous People, National Changhua University of Education, Changhua 50007, Taiwan; oscarhong@cc.ncue.edu.tw

**Keywords:** homosexual long-term care, older adults male homosexuals, long-term care services

## Abstract

As the significance of long-term care services for older adults becomes more pronounced in Taiwan, especially considering the intensifying trend of an aging population, there remains a lack of comprehensive attention to the long-term care needs and experiences of older adult individuals within the LGBTQ+ community. The present study examines the long-term care service awareness, needs, and usage intention of older adult male homosexuals in Taiwan and assesses their ideal long-term care service model. This study of five older adult male homosexual subjects aged 66 to 73 years is intended as a preliminary exploration. Interviews were used to collect data. The study determined that the five subjects displayed high awareness of long-term care services, as they possessed actual experience of these services or had even participated in care service staff training to obtain certificates. Some even had experience in applying for home care services and reported problems during use. The subjects perceived that they were very likely to require long-term care services in the future and tended to opt for home care services if they required long-term support. Due to their personal experiences, the subjects displayed negative awareness of long-term care services and expressed worry that long-term care service staff harbored poor attitudes toward homosexuals. The subjects considered the most important aspects of long-term care to be basic medical care and lifestyle care. However, they worried that long-term care staff would delay or refuse to provide such services due to the subjects’ sexual orientation or stereotypes related to it, and they were concerned above all about the “friendly attitude” of long-term care staff. They hoped that long-term care staff were friendly toward homosexuals and did not discriminate against them, feeling that it would be more appropriate for homosexual long-term care staff to provide assistance. In terms of vision, while preferring organizations with homosexual employees, the subjects worried that they would be stigmatized and discriminated against. Regarding ideal long-term care services, while considering institutions with homosexual staff to be ideal, the subjects also worried that these would be labeled as institutions that were dedicated to homosexuals, potentially resulting in discrimination. Therefore, they hoped that the sexuality sensitivity of long-term care staff could be improved and that they would undergo professional continuing education to learn about homosexuals, their situations, and care needs.

## 1. Introduction

Taiwan holds the distinction of being the first country in Asia to legalize same-sex marriage. This landmark decision significantly boosted the visibility of the lesbian, gay, bisexual, transgender, and queer (LGBTQ+) community nationally. With this increased recognition, all sectors in Taiwan, including long-term care, must now address relevant LGBTQ+ issues. This necessitates modifying services to accommodate distinct needs and ensure adequate support for LGBTQ+ individuals [1].

Taiwan is also among the countries experiencing the fastest population aging globally [2]. To tackle the challenges of an aging population and a declining birth rate, the Taiwanese government has promoted long-term care as an essential issue on the policy agenda, actively addressing the emerging needs and issues related to an aging society. Nevertheless, the government’s focus on long-term care services for the LGBTQ+ community remains relatively limited [3]. The presence and specific needs of LGBTQ+ individuals in the realm of long-term care have largely been overlooked and rendered invisible [4]. What research exists on the needs of older adults rarely addresses the unique circumstances of the LGBTQ+ population. Older LGBTQ+ individuals often lack confidence in the friendliness and acceptance of service providers, leading to lower utilization of care and medical services. Compared to their heterosexual counterparts, older LGBTQ+ individuals display a greater need for care [5]. Their frequent concealment of identity, driven by fears of discrimination, ridicule, or harassment, can seriously harm their mental and physical health. This concealment and its consequent health impacts significantly increase the chances of requiring long-term care [6]. Many older LGBTQ+ individuals feel fearful when considering entering long-term care facilities [7]. Such fears can lead to psychological distress, including suicidal thoughts [8,9,10]. Older LGBTQ+ individuals worry about being subjected to institutional control, prejudice, and discrimination. Consequently, they often opt not to disclose their sexual orientation and gender identity, making them virtually invisible in long-term care settings [11]. Their invisibility also means they are overlooked in gerontology, policymaking, and law [12]. Therefore, a general lack of awareness about LGBTQ+ needs in long-term care often leads long-term care workers to presume heterosexuality in all their clients [13,14].

Hong indicated that middle-aged gay men exhibit different long-term care service needs compared to heterosexual individuals. Efforts to address LGBTQ+ long-term care needs may face unique challenges distinct from those of the heterosexual population, including the following issues. 1. Social discrimination and prejudice: LGBTQ+ individuals often face societal discrimination and prejudice, potentially impacting their experiences in medical and care facilities. They may be more cautious or concerned about revealing their sexual orientation or gender identity in long-term care environments. 2. Family structure: the family structure of LGBTQ+ individuals may differ from that of traditional heterosexual family structures, involving same-sex partners, adopted children, or other family forms. In long-term care, considerations for such family structures are essential if we are to ensure adequate support and respect. 3. Social support networks: the LGBTQ+ community may possess unique social support networks that play a crucial role in long-term care. Care institutions should be aware of these networks and create an environment that is friendly and inclusive for LGBTQ+ older adult individuals. 4. Gender medical needs: some LGBTQ+ individuals may undergo gender-related medical procedures, such as gender confirmation surgery. Providing appropriate medical support and understanding is crucial to meeting these specific health needs. 5. Mental health: due to societal pressures and discrimination, LGBTQ+ individuals may face mental health challenges. In long-term care, it is important to offer appropriate psychosocial support and mental health services. 6. Legal rights: legal protections for LGBTQ+ rights vary across different regions. Ensuring respect for and protection of the legal rights of LGBTQ+ individuals in long-term care settings is crucial. In summary, creating a long-term care environment that is LGBTQ+-friendly, inclusive, and respectful of their unique needs is key to ensuring proper care for all older adult individuals. This requires efforts to educate healthcare professionals, formulate relevant policies, and establish culturally sensitive medical and care institutions. Due to the lack of fundamental social protections and a prevailing homophobic atmosphere in society, LGBTQ+ individuals may develop a sense of self-defense, leading to common experiences of concealing their sexual orientation or gender identity to avoid rejection and exclusion. Other studies indicate that older adult homosexuals tend to avoid or delay seeking medical services, primarily due to discrimination based on their sexual orientation and gender identity by healthcare providers and social service personnel. The research also highlights that the choices for informal care for older adult homosexuals are decreasing. Older adult homosexuals are more likely to be single or living alone, and they are unlikely to have children to care for them. They often rely on family-like networks composed of friends and LGBTQ+ social welfare organizations for caregiving [15,16].

The assumption of heterosexuality can make LGBTQ+ individuals feel that their identity is misunderstood and that their partners and friends are neglected in care-planning and decision-making processes [17,18]. In seeking services, older LGBTQ+ individuals and their caregivers may confront extra stress, burdened with concerns about acceptance, the necessity to conceal their identities, respect, and safety [19]. Due to fears of discrimination or culturally incompetent care, LGBTQ+ individuals often avoid using formal care services [6,20]. A total of 92% of older LGBTQ+ individuals have indicated that acceptance is a crucial factor in choosing care services [21]. Research has examined the hopes and fears of older LGBTQ+ individuals contemplating long-term care at the end of their lives, identifying prevalent worries. These include social isolation, reduced autonomy, heightened vulnerability to internalized homophobia, and the risk of being in unsafe environments [22]. According to demographic studies, older LGBTQ+ individuals tend to display inferior physical and mental health compared to their heterosexual peers [23].

As outlined above, this study aims to provide insight into the perceptions, needs, and willingness of five older GBM to use long-term care services and understand their ideal vision for such services.

## 2. Methods

### 2.1. Design

This study, initiated in 2015, aimed to explore long-term care issues within Taiwan’s LGBTQ+ community. Initially focusing on older LGBTQ+ individuals, we encountered challenges as no participants from this group were identified. The literature suggests that, due to pressures from heterosexual cultural norms, older LGBTQ+ individuals may be less willing to share their experiences openly. The research shifted its focus to middle-aged GBM, subsequently surveying 202 LGBTQ+ participants during the Taiwan Pride Parade in 2017 to raise societal awareness about long-term care. In 2023, we organized the “Taiwan LGBTQ+ Long-Term Care Exhibition”, where encounters with GBM engaged in home care inspired a more in-depth investigation.

This study was exploratory. Given the limited existing literature in Taiwan on the topic, exploratory research was performed to gain an in-depth understanding of a relatively unfamiliar field. Exploratory research provided foundational information, assisting in opening new research avenues, offering preliminary insights, and laying the groundwork for further studies. To conduct this research, in-depth interviews were chosen as the method for data collection. Qualitative research can offer preliminary direction and theoretical framework for topics that have not been extensively explored. [24]. Given the lack of related research in Taiwan and Asia addressing the stated objectives of this study, qualitative in-depth interviews were utilized for exploration.

### 2.2. Participants

This study involves five older LGBTQ+ individuals as participants, with an average age of 69 years (Table 1).

### 2.3. Data Collection

When conducting the research, this study employed purposive sampling to ensure the diversity of the study participants, aiming for a comprehensive understanding of the perceptions, needs, willingness to use, and ideal long-term care services among older adult gay men. Through purposive sampling, we specifically selected cases that aligned with the research theme, allowing for an in-depth and holistic exploration of the perspectives and experiences within this particular demographic. This sampling strategy contributed to providing more representative and nuanced data, enhancing the credibility and applicability of the research findings. The study employed a semi-structured interview outline, developed by the researchers in 2018, to guide the interview process. Interviews were the primary method of data collection. Participants were asked for their consent before setting a date for the interviews, and confirmation calls were made on the day of the interview to finalize the location and time. Digital recording devices were used throughout to capture the content of the interviews (Table 2).

### 2.4. Data Analysis

We adopted thematic analysis to assess the study data. The interview transcripts were coded, and through this coding process the categories and themes relevant to the research objectives were progressively extracted from the interview data. The data collected underwent analysis using qualitative content analysis, as outlined by Graneheim and Lundman [26]. This study adhered to the following steps: (1) The interview transcripts were meticulously read multiple times by the interviewee to establish a comprehensive grasp of the overall content. (2) The textual data were read and systematically coded, a process involving continual cross-referencing and comparison to unravel meanings and relationships within the data. (3) Inductive analysis was applied to the data to uncover shared themes, following which we classified data with shared meanings and constructed core categories and subcategories. (4) Meanings, patterns, and concepts were extracted from the collected data, ultimately leading to the formulation of the study’s findings.

### 2.5. Research Ethics

In order to respect and protect the rights and privacy of the participants, consent forms detailing the study’s purpose, process, and ethics were prepared. Participants were required to read these forms thoroughly, enabling them to understand the purpose of the study, its procedures, and the time commitment involved. Further, it allowed them to understand their right to participate or withdraw at any point voluntarily and the principle of confidentiality. All these ethical considerations were aimed to minimize any risks. Interviews were conducted only after reaching a consensus with the participants. The identities of the five study participants were anonymized using codes to protect their privacy.

## 3. Results

The analysis of the study results is divided into four subsections. Section 3.1 describes the understanding of long-term care services among older GBM in Taiwan; Section 3.2 summarizes their needs regarding these services; Section 3.3 presents their willingness to utilize long-term care services; and Section 3.4 discusses ideal long-term care services from the perspective of older GBM in Taiwan.

### 3.1. Understanding of Long-Term Care Services among Older Gay Bisexual Men in Taiwan

The study indicated a high level of awareness about long-term care services among older GBM, explicable due to their prior experiences with the services. They were well informed about the contents of long-term care services.


*“My mother was bedridden for a long time after a fall. Back then, our financial situation allowed us to hire a caregiver. After spending significantly over the years, my brother, who was opposed to this, insisted on moving her to a nursing home. I disagreed and chose early retirement to care for my mother myself. This experience made me familiar with various long-term care services and what’s available.”*
(PD)


*“I enrolled in a healthcare aide training course to better understand the fine details needed in older adults care and learn about taking care of my health as I age. It also helped me appreciate the hard work of healthcare aides so that I would not be unreasonable and demanding when I am under care. I was delighted to receive my healthcare aide certificate. Initially, I wanted to work as a healthcare aide, but my relatives and friends opposed it, citing my age and the after-effects of a car accident. My partner works in home care; thus, we discuss current care services, and I am clear about the subsidies available for long-term care services.”*
(PA)


*“I was hospitalized for prostate cancer surgery and applied for home care afterward. My understanding of care services improved thanks to a friend who works in home care and has been a great help... He helped set up an accessible space in my house and found a gay home care aide. I cannot go to work now and work from home, with a gay home care aide assisting me with daily tasks.”*
(PC)


*“I have experience with it since my family members and relatives have applied for home care or hired foreign caregivers. I understand that poor health can lead to many inconveniences; even bathing and eating require help. Therefore, I try my best to stay healthy and exercise as much as possible.”*
(PE)

When there was a need for long-term care services, older GBM, due to their personal experiences, often hoped to be treated by long-term care workers who were themselves members of the LGBTQ+ community.


*“Diagnosed with this cancer, I initially felt hopeless. During a previous hospital stay, a lady was caring for me, which was awkward and made me feel very sad. Fortunately, a friend understood my mental strain and reminded me that death is an inevitable part of life, encountered at any age. He advised me to face it positively and to undergo active treatment as long as my physical condition allowed. He then helped me find a gay home care aide. I find that care from a gay person is less embarrassing. We can talk about topics like gay senior living and emotional life.”*
(PC)


*“If the care provider is also gay, there is a better understanding. Specifically, we do not need to hide anything. We are all gay; there are no awkward barriers.”*
(PB)


*“Previously, my partner, a home care aide, took care of me. Thus, I believe care services from a gay are better. I have seen many home care aides who are older women, chatty and nosy. If they discovered something about me, the gossip would be endless.”*
(PA)

When questioned regarding the types and functions of long-term care services, five older GBM were clear about these areas due to their deep engagement with such issues. For instance, PC, who had applied for home care, knew precisely which services he could use, and PA, having taken a healthcare aide course, was also very familiar with long-term care. The other participants were acquainted with these services because their parents had used them.


*“I took a healthcare aide course. Therefore, I understood it well and encouraged my partner to enroll. He now has more stable work because of it.”*
(PA)


*“My mother stayed in a care facility, so I understand institutional care. My father attended daycare services; thus, I am also familiar with that.”*
(PE)

All five older GBM exhibited similar levels of awareness and understanding towards long-term care workers but harbored negative perceptions, worrying about their potent attitudes towards LGBTQ+ clients. Their concerns stemmed from personal experiences.


*“In the healthcare aide course I attended, many students were older adult women who do not necessarily accept gay individuals. When my partner attended a continuing education class where homosexuality was mentioned, the women showed no interest, and he also avoided coming out at work, fearing discrimination.”*
(PA)


*“From my observation, long-term care workers are not very gay-friendly… In that environment, even older clients are not treated with much respect. Some heads of long-term care facilities are arrogant, making it evident that coming out is not an option.”*
(PC)


*“I have always encountered unfriendly attitudes toward gays, and it is still the case. The older they are, the less they understand gays, and it is the same with long-term care workers now; they hardly know any gay individuals.”*
(PD)

All five older GBM expressed a wish for long-term care workers to be LGBTQ+-friendly and non-discriminatory, suggesting that care from LGBTQ+ long-term care workers would be more fitting and comfortable.

### 3.2. Need for Long-Term Care Services among Older Gay Bisexual Men in Taiwan

When questioned about the likelihood of requiring long-term care services, all five older GBM indicated a high probability. Furthermore, all cited prior negative experiences with such services.


*“Even if I had no prior experience, I would still feel the need. Everyone, at some point, requires care. In our family, some relatives live in facilities, and foreign domestic workers take care of others, so the need is universal.”*
(PD)


*“The need is inevitable, and early learning of caregiving knowledge and skills is crucial. At any age, as long as one is physically able, I encourage everyone to attend healthcare aide courses. Working as a healthcare aide is not mandatory, but at the very least, it helps increase medical knowledge and self-care skills, which can be handy when family members need care. In the event of an accident or disability, I would need long-term care services. Home or institutional care depends on government policies, regulations, and financial capability.”*
(PA)


*“Of course, I would need long-term care services if I become bedridden or develop dementia without hope of recovery. I fear that the government’s eligibility criteria for applying for these services are so stringent that they are practically inaccessible and merely a tool for political campaigns.”*
(PD)


*“I do need it, but I try my best to maintain my health to avoid being bedridden and dependent on others.”*
(PB)

All five older GBM believed that support from friends and partners in caregiving was critical. They were less likely to receive care from family members due to systemic societal discrimination and a lack of a public care system. Some LGBTQ+ individuals also refrained from seeking help from their families due to not having come out to them.


*“In my understanding, caregiving responsibility naturally falls to the family. Friends and partners are not obliged to care; they only provide moral support. Without family, one has to rely on friends, which depends on how close these friendships are. It is all relative—you get back what you put in, depending on the people you interact with. In seeking help, I would turn to my partner, as he has a healthcare aide certificate. Of course, he should be paid for his services. Further, some long-term care workers or facilities worry about the sexual needs of gay clients. In reality, most gays who require long-term care no longer have sexual needs; rather, the concern should be about protection from sexual harassment. In our times, gays could only seek relationships and live their lives in the shadows; thus, our thoughts tend to be more pessimistic, and hoping for a peaceful end is a blessing.”*
(PA)


*“Friends often provide more care, especially for older gay men who lack a partner, their own housing, or savings. They need a facility that offers shelter and three meals a day. A model similar to some charitable organizations could be considered, where those still capable of manual dexterity can engage in simple manufacturing within the facility to help supplement their income.”*
(PD)

Additionally, the most crucial requirement was equal care. All five older GBM believed that the most essential needs in long-term care were primary medical and daily life care. However, there was apprehension as to whether long-term care workers might delay or refuse services due to sexual orientation or stereotypical misconceptions. The “friendly attitude” of the long-term care workers was their primary concern.


*“I once hired a lady who made it very awkward for me. She would ask why I, at my age and with my earning capacity, lived alone and was not married. I had to brush off her questions casually. She did not know I was gay. So, there were no serious issues, but it was tiring to dodge these questions in care. I find it easier to chat with gay home care aides; there is less judgment and more common ground in conversations.”*
(PC)


*“It would be even better if male gay healthcare aides who are careful and skillful, especially in physically demanding services, were available. They tend to be more competent and comfortable with female care recipients, avoiding awkwardness. Similarly, male recipients have better psychological comfort and less discomfort.”*
(PA)


*“If they could empathize with the social circumstances of gays, that would be preferable. However, currently, there is a lack of education that allows long-term care providers to understand and recognize gays.”*
(PD)


*“As I just mentioned, there is little awareness about gays in long-term care. If they could learn to interact with gays in a friendly and affirmative manner, perhaps many would not be as worried. Many gay individuals I know are concerned about being bullied or having to lie about their preferred gender if they end up in a nursing home.”*
(PE)

### 3.3. Willingness to Use Long-Term Care Services among Older Gay Bisexual Men in Taiwan

When queried about the perceived likelihood of using long-term care services, all five older GBM indicated a high probability, primarily due to their experiences with such services.


*”We predominantly favor home-based care. In my case, I have already applied for home care services. Financially, I am pretty sound and can manage costs over 30,000 NT$ monthly. Nonetheless, my primary concern is that my current illness might not quickly subside.”*
(PC)


*”I prefer getting care at home. In the future, I am considering the option of hiring foreign caregivers.”*
(PE)


*“I had a disheartening experience when I applied for a veterans’ home. As soon as they found out about my sexual orientation, they immediately rejected me. That incident made me quite angry. I believe it is better to stay physically fit and go for home care instead of residing in an institutional setting.”*
(PB)


*“My choice is home care, mainly because my partner can care for me, contributing to his income.”*
(PA)


*“I tend to visit local community care centers when my health allows. Otherwise, I will have to rely on home care services.”*
(PD)

The group noted that the average cost for self-funded long-term care services was over NTD 40,000.


*“Economically, I am in a position where I can afford a bit over 40,000 NT$ each month for these services.”*
(PC)


*“Considering the care expenses it is generally about 40,000 NT$ or more. Since many gay men are unmarried and without family support, it is crucial to evaluate their capacity to shoulder such financial burdens. Generally, those requiring long-term care cannot work, leading to minimal or no income.”*
(PE)


*“The decision to apply for long-term care depends on one’s health situation and financial means. My retirement pension is just over 30,000 NT$. I can only afford up to 1000 NT$ daily for care services. When factoring in medical costs, it appears that I have no alternative but to await death at home. The expenses for caring for my mother amounted to over 60,000 NT$ monthly, nearing a million annually. Over the seven years we employed a foreign caregiver until her passing, the total expenditure was approximately seven million NT$.”*
(PA)

### 3.4. Ideal Long-Term Care Services from the Perspective of Older Gay Bisexual Men in Taiwan

Older LGBTQ+ individuals agree on the concept of constructing LGBTQ+-specific long-term care facilities.


*“I dream of an environment akin to a villa with a garden, accessible facilities, meals, hygiene care, and all at an affordable cost. Ideally, these should be gay-specific systems designed by and for gays, whether male or female. If heterosexual norms dictate the standards, they are simply outdated.”*
(PD)


*“A gay-specific institution would be preferable, but its regulations must be determined privately, not widely broadcast. If everyone knows it is a gay long-term care center, residents might face pointing fingers whenever they step outside.”*
(PE)


*“The best would be an institution for gays, but it might get stigmatized as “exclusively for gays,” inviting different treatment... Government regulations stipulate specific requirements for establishing long-term care facilities in terms of equipment and zones. Therefore, we should check whether these government regulations align with the realities of society and urge the government to make improvements. My preference for a specific type of environment is not the main issue. I once aspired to create a gay-friendly daycare service. I bought a ground-floor shop, considering its accessibility for older adults with mobility issues and easier evacuation in emergencies. When I inquired about its establishment, my application was rejected because commercial areas are not permissible locations, dousing my plans in cold water. The permissible areas are upstairs in residential zones or independent houses with bathrooms, layouts, and corridors that must all be equipped with accessible facilities and space.”*
(PA)


*“I am aware of a gay-friendly elder care association managed by a gay. It remains to be seen whether they will evolve into a gay-specific long-term care facility. However, if they do, I would be willing to move in. Being surrounded by fellow gay individuals would undoubtedly offer a sense of reassurance. We could also learn from Japan’s gay-friendly senior housing. These places are staffed by managers who assist with shopping, medical appointments, and other basic life needs. Particularly appealing are the low rents communal dining areas that encourage interaction among residents, thus preventing isolation. Forming friendships here and mutually aiding each other can enhance independent living. Many single elders face difficulties renting homes because landlords fear what might happen if something goes wrong. Older gays are likely to encounter these issues as well.”*
(PB)

The unanimous opinion among the five older GBM interviewed is that an increase in gender sensitivity among long-term care workers is imperative, in addition to the development of LGBTQ+-specific long-term care facilities.


*“Our biggest concern is the perspective of the long-term care workers towards us. If they exhibit homophobia, whether they live in a facility or use home care services, it would be a source of immense distress. Therefore, care providers must be knowledgeable about gay individuals and understand the social predicaments faced by the gay community and the specific care challenges we encounter. They must be able to see things from our perspective and comprehend our needs.”*
(PD)


*“There is a definite need to enhance the understanding of gay issues. My partner mentioned that continuous education on gender sensitivity for long-term care workers rarely addresses gay topics. They cover other gender issues instead. How can care providers comprehend our problems without being educated about them? How can we dare to come out of the closet? Unless care providers first understand what being gay means and the concerns gays have regarding long-term care, we are too frightened to reveal our true selves. We have to remain hidden because our generation has been hurt too much in the past. In our times, gays could only seek affection while hiding in the shadows. No wonder our outlook is rather bleak and pessimistic. To die well is a fortune we hope for at the end.”*
(PA)


*“Most of the older gays I know are retired and in relatively good health, but we still contemplate future care for one another. As friends age, who will take care of whom? That is why we discuss the importance of having long-term care providers who understand and do not discriminate against gays. If they are knowledgeable and nonjudgmental, we will consider living in care facilities and utilizing daycare and home care services.”*
(PB)


*“Living to this age, one starts to consider life in old age. We have insurance to assist us, but human care is unpredictable. The news shows only a fraction of the mistreatment that happens. Much remains unseen. I have heard from social workers in long-term care facilities that older adults are not always treated kindly, and as gays, that scares us even more.”*
(PE)


*“I have asked my home care aide, who said they must take courses on gender sensitivity. Nevertheless, these courses seldom cover the care of gay individuals. It is disappointing because if instructors do not address the reality of being gay, the care providers’ understanding of gays will be based solely on what they see on TV. They will not realize that there are many types of gay individuals, not just the stereotypes portrayed on television. Therefore, it is vital to elevate the gender sensitivity of care providers and equally important to educate them on the care of the gay population.”*
(PC)

## 4. Discussion

The analysis presented herein reveals that the five older GBM display considerable awareness of long-term care services. This understanding is rooted in actual experience, both in directly utilizing these services and dealing with the attendant issues. All participants acknowledge a high probability of future dependence on long-term care services. Should the need arise, they display a preference for home care services. Hong’s study on middle-aged GBM similarly found similar a tendency toward utilizing home-based care services [25]. They are also conscious of the looming issue of long-term care, with society confronted by a scarcity of personnel, and some have participated in caregiver courses and wish to engage in gay-friendly custodial services. In contrast to heterosexuals, GBM have more significant concerns stemming from social safety nets, pension schemes, and issues related to health and long-term care as they enter old age [20,27]. Older LGBTQ+ individuals agree on the concept of constructing LGBTQ+-specific long-term care facilities. This idea mirrors the vision shared in Hong’s interviews with middle-aged LGBTQ+ people, highlighting the desire for such dedicated institutions. However, there is also a fear of being labeled [28].

Consequently, older GBM have a heightened awareness of care services, informed by their own experiences and those of their close friends and family who have utilized such services. However, Mark et al. illustrated that studies on the needs of older adult rarely address the unique circumstances of the LGBTQ+ population, and that the apprehension of this demographic stems from personal experiences regarding the attitude of caregivers toward LGBTQ+ individuals [4]. Shauna identified that older LGBTQ+ individuals often face discrimination and bias in long-term care and are frequently compelled to confront their sexual orientation during the aging process [29]. Hence, men in this cohort believe that, when entering long-term care beyond basic medical and life care, their primary concern is whether caregivers might refuse or delay services due to sexual orientation or stereotypical misconceptions. The friendly attitude of long-term care workers is of paramount importance to them. Bridget and Lisa suggest that the risk of needing long-term care for older LGBTQ+ individuals is higher than that for heterosexual elders [30]. Many older LGBTQ+ individuals feel fear at the prospect of entering long-term care facilities [21]. Research in Taiwan by Wang [31] indicates that older GBM face dual social exclusion as they are both invisible in heterosexual society and unwelcome within the LGBTQ+ community due to their aging bodies. Gofyy voices the experience of Taiwan’s older GBM who have lived through the stigma of homosexuality being deemed “dirty, dark, and shameful,” a stain many still struggle to erase [32]. Having experienced such stigma, the five older in this study GBM hope for long-term care workers who are gay-friendly and non-discriminatory, suggesting that care providers who are themselves members of the LGBTQ+ community might offer more pertinent support. While they see the value in dedicated LGBTQ+ facilities, they worry about being stigmatized or viewed differently, aligning with the desires and concerns expressed by Hong [33] for exclusive LGBTQ+ long-term care facilities without the fear of labeling.

In addition, LGBTQ+ individuals typically prefer to seek assistance from friends and partners rather than from their biological families or the health and welfare system [34,35]. These men advocate for improved gender sensitivity among long-term care workers and call for continuing education that recognizes LGBTQ+ people, their situations, and their specific care needs. Choi and Meyer found that many studies show older LGBTQ+ individuals to avoid or delay medical services due to discrimination during long-term care, with many also possibly living alone or in solitude [36]. Access to supportive housing in later life is a critical concern for older LGBTQ+ individuals, yet most long-term care providers are not adequately prepared to meet their needs [37]. Older GBM bear significant pressures and experience a heightened sense of repression compared to their younger counterparts. Some are still shackled by the stigma of their past, leading to a perpetual self-scrutiny marked by negative perceptions such as “I am not accepted” or “the other person dislikes me.” For older members of the LGBTQ+ community, various social conditions often present disadvantages that adversely affect their health. Older patients suffer from age discrimination by healthcare professionals when using health services. Ageism by healthcare professionals influences the health service utilization rates of older people. In fact, older people often exhibit a negative self-perception of their aging process, linked to internalized stereotypes and ageism by healthcare professionals. This latter issue can cause an imbalance in power relations that undermines older patients’ dignity, needs, and opportunities [38]. According to a 2018 AARP survey, over 60% of LGBTQ+ individuals aged 45 and above harbored concerns about potential neglect, abuse, harassment, or restricted access to services in long-term care facilities; alarmingly, 54% of gender-diverse respondents feared that concealing their identity was a prerequisite for accessing suitable care centers [39]. Additionally, research by SAGE and the Equal Rights Center indicated that up to 48% of older LGBTQ+ individuals have faced discrimination when seeking long-term care services.

Although same-sex marriage has been legalized in Taiwan, heterosexual norms persist within the culture, implying an inherent societal expectation of heterosexuality and rendering other gender diversities as ‘unnatural’, thereby perpetuating a culture of heteronormativity. The question of whether long-term care in Taiwan can respect and accommodate the needs of the LGBTQ+ community remains pertinent for older LGBTQ+ individuals. Taiwanese society’s insufficient support system for older LGBTQ+ persons increases the severity of their care challenges. Therefore, it is crucial for long-term care workers and facilities to cultivate an understanding and recognition of the unique social and psychological experiences of LGBTQ+ individuals. Through education tailored to LGBTQ+ issues, both can enhance their empathy and comprehension of the LGBTQ+ plight, fostering an environment where values of inclusivity and attitudes of friendliness prevail. This would enable the delivery of diverse professional services to be more effective in terms of collaboration and specialization.

## 5. Conclusions

The study determined that the five subjects displayed high awareness of long-term care services, as they possessed actual experience of these services or had even participated in care service staff training to obtain certificates. Some even had experience in applying for home care services and reported problems during use. The subjects perceived that they were very likely to require long-term care services in the future and tended to opt for home care services if they required long-term support. Due to their personal experiences, the subjects displayed negative awareness of long-term care services and expressed worry that long-term care service staff harbored poor attitudes toward homosexuals. The subjects considered the most important aspects of long-term care to be basic medical care and lifestyle care. However, they worried that long-term care staff would delay or refuse to provide such services due to the subjects’ sexual orientation or stereotypes related to it, and they were concerned above all about the “friendly attitude” of long-term care staff. They hoped that long-term care staff were friendly toward homosexuals and did not discriminate against them, feeling that it would be more appropriate for homosexual long-term care staff to provide assistance. In terms of vision, while preferring organizations with homosexual employees, the subjects worried that they would be stigmatized and discriminated against. Regarding ideal long-term care services, while considering institutions with homosexual staff to be ideal, the subjects also worried that these would be labeled as institutions that were dedicated to homosexuals, potentially resulting in discrimination. Therefore, they hoped that the sexuality sensitivity of long-term care staff could be improved and that they would undergo professional continuing education to learn about homosexuals, their situations, and care needs. Based on the findings of this study, the following recommendations are proposed: 1. Filling research gaps: there is a need to provide a comprehensive understanding of the care requirements of the older adult gay men community, addressing existing research gaps. 2. Personalized services: researchers must aid in the establishment of long-term care service models that better align with the actual needs of older adult gay men, thereby enhancing the effectiveness of care. 3. Promoting inclusive environments: public health must advocate for the creation of more friendly and inclusive care environments for older adult gay men. 4. Policy recommendations: scholarship may offer insights that allow the government to formulate more targeted policies, supporting the long-term care rights of older adult gay men. 5. Raising social awareness: we must in-crease societal awareness of the challenges faced by older adult gay men in long-term care, fostering broader social understanding. Additionally, our research group has established the “LGBTQ+-Friendly Long-Term Care” project in Taiwan. Through research, parades, comics, exhibitions, lectures, and education, we aim to promote equality in Taiwan, ensuring that everyone can age with dignity. Our goal is to eliminate differential treatment and discriminatory practices in Taiwan’s long-term care services, eliminating exclusion based on gender, sexual orientation, gender identity, marital status, age, physical or mental disabilities, illnesses, social class, race, religious beliefs, nationality, or place of residence of the service recipients. Despite the contributions made, our study has limitations, including the following concerns. 1. Sample limitations: the participants may not fully represent the entire older adult gay men community in Taiwan, potentially limiting the generalizability of the study results. 2. Social–cultural Influences: the long-term care needs and experiences of Taiwan’s older adult gay men are influenced by social–cultural factors, which may vary across different regions, ethnic groups, or socioeconomic strata, limiting the study’s applicability. 3. Subjective evaluation: the use of qualitative methods, such as in-depth interviews, may introduce subjective biases from researchers, potentially affecting the comprehensive reflection of participants’ true circumstances.

## Figures and Tables

**Table 1 healthcare-12-00418-t001:** Basic information of study participants.

Participant	Age	Education	Marital Status	Relationship Status	Coming out Status	Profession	Belief
A	71	College	Married but separated from opposite-sex partner	In a same-sex relationship	Out to friends, not to family	Retirement	Buddhism
B	73	Junior high school	Never married	Single	Out to family and friends	Retirement	None
C	69	College	Never married	In a same-sex relationship	Out to friends, not to family	Government employee	Taoism
D	67	Master’s	Never married	Single	Out to friends, not to family	Lawyer	None
E	66	Senior high school	Divorced	In a same-sex relationship	Out to family and friends	Construction worker	Taoism

Source of data: produced by researchers.

**Table 2 healthcare-12-00418-t002:** Interview guide.

Facet	Interview Guide
Perception of long-term care services	From the past until now, what is your perception or understanding of long-term care services? From the past until now, what is your perception or impression of long-term care personnel? If you have a need for long-term care services, would the gender of the long-term care personnel affect your decision to seek or use these services? If yes, what are the reasons? If not, what are the reasons? How do you obtain information related to long-term care services? What is the content of the information? Have you ever discussed issues related to long-term care services with others? If so, who were the discussion partners (e.g., friends, partners, family, or others)? What was the frequency and duration of these discussions? If you have not discussed these issues with others, what are the reasons?
Demand for long-term care services	Do you perceive the possibility of needing long-term care services in the future (e.g., the potential need for long-term care services due to disability, dementia, or mobility issues associated with aging)? In the caregiving process, whom do you think will bear the responsibility of care? (e.g., friends, partners, family, etc.) In the future, if you were to face disability, dementia, or mobility challenges due to aging and require care, who do you envision as the primary source of assistance (e.g., friends, partners, family, or others)? In the future, if you were to face disability, dementia, or mobility challenges due to aging and require care, what channels would you seek to find resources? What long-term care service needs do you anticipate having in the future (e.g., health, daily living, financial, etc.)? In your opinion, which aspect of long-term care services do you believe is the most challenging or requires the most assistance for individuals in the LGBTQ+ community?
Willingness to use long-term care services	In the future, if you age or encounter unexpected situations (such as illness), do you envision using long-term care services? If yes, what are the reasons? If not, what are the reasons? If you anticipate a need for long-term care services in the future, which service model do you think you would prefer? (The preferred way of receiving the needed long-term care services.) What are the reasons for this preference? Would you be willing to pay for long-term care services out of pocket? If yes, how much would you be willing to spend on long-term care services? If not, what are the reasons?
Ideal long-term care services	How do you envision long-term care services that align with your ideal? Please provide a description (e.g., environment, services, facilities, etc.). If a dedicated long-term care system specifically for the LGBTQ+ community were established (e.g., institutions, services, personnel, etc.), how would you feel about it? What are your views and level of acceptance? Do you have any thoughts or suggestions regarding long-term care issues for the LGBTQ+ community?

Source of Data: Produced by [25].

## Data Availability

All data included in this study are available upon request by contacting the corresponding author.

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
