# Peer review of "Study on Long-Term Care Service Awareness, Needs, and Usage Intention of Older Adult Male Homosexuals in Taiwan and Their Ideal Long-Term Care Service Model"

_healthcare, 2024, doi:10.3390/healthcare12040418_

Round 1
Reviewer 1 Report
Comments and Suggestions for Authors
1. Abstract: Long-term care involves providing services to individuals with disabilities. Regardless of whether one receives or provides long-term care services, the assessment criteria do not have a sexual orientation. The abstract suggests that interviewed individuals, both recipients and providers of long-term care services, disclose their sexual orientation. What are the reasons behind this disclosure?
2. Introduction: Lines 43-79 repeatedly emphasize the overlooked long-term care needs of the LGBTQ+ community. The author is requested to clarify the statement regarding the long-term care needs of LGBTQ+ individuals and how it differs from the heterosexual community.
3. Lines 80-90: The laws passed in the United States focus on elderly individuals who are HIV positive. According to Taiwanese law, HIV positivity must be reported to the local health department. If a case involves an individual who is HIV positive, both recipients and providers of long-term care services have the responsibility to disclose this information.
4. Question regarding the HIV status of the 5 interviewed GBM cases: Could the author provide the HIV status of the 5 interviewed GBM cases?
5. Lines 105-114: The author conducted interviews with 5 GBM cases in 2021, while the IRB approval date is in 2023. This violates academic ethics: IRB approval must be obtained before conducting case interviews.
Author Response
Dear reviewer,
Thank you very much for your guidance, it has been invaluable in my learning process. I have incorporated the suggestions you provided into my work, as shown in the PDF below.

Reviewer 2 Report
Comments and Suggestions for Authors
Firstly, I express my gratitude to the journal for the opportunity to review this manuscript, which delves into crucial issues surrounding the perceptions, needs, and willingness of five older GBM to use long-term care services and comprehend their ideal vision for such services. I commend the significant effort invested in this research, mainly due to its focus on a vulnerable population. Additionally, I underscore and highlight the social justice orientation of this study. My intent with the following comments is to propose enhancements to this compelling research work, supplementing the feedback from my fellow reviewers. On another note, refining these major topics would enhance the overall quality of your research.
TITLE AND ABSTRACT
- Please consider using the term “older people” or “older adults” instead of “elderly”, thus following the best practice guidelines from the new editions of the AMA, APA, AP, and AGS style guides for use when writing about older adults.
- I suggest introducing some background lines in the first part of the abstract before the aims and a brief conclusion followed by a few lines regarding clinical/scientific contributions of the research.
INTRODUCTION
- Please relocate lines 91-114 from the introduction to the methods section, summarised in a few lines. This part relates to the origin of this study, not to any background/ introduction of scientific literature that could contribute to contextualising your research.
METHODS
- I suggest changing the subsections “Research Methods” to “Design” and “Research Tools” to “Data Collection”.
- Please eliminate lines 120-121. They appear already precisely the same at the end of the introduction section.
- Line 122: “In-depth interviews were chosen as the research method to achieve this.”: interviews are not a method but rather a data collection tool. You mentioned that this is an exploratory study, but do you mean, therefore, a phenomenology? If it is so, is it a descriptive or interpretative one? Please justify the method´s election for this study.
- Lines 130-131: “(…) all participants were male, ranging in age from 66 to 73 years, with three being retired and two still employed.” Readership can actually see age ranging in Table 1. Report instead the age mean in the text.
- How was the sampling method? Was it purposive, by convenience…? Please state this with a more detailed recruitment strategy within the “Research Tools/ Data Collection” section.
- Concerning the “Data Analysis” section, were more than one researcher involved in it? If so, did they have meetings to discuss the analysis phases? You say you employed a coding process but do not add any information about this. How was the analytical coding phase conducted?
RESULTS
- Please properly distinguish verbatims from narrative developments in your results. For this, I recommend placing quotes from participants in quotation marks and reducing the font size by one point compared to the rest of the manuscript.
- I suggest using no more than two to three quotes from participants to enlighten the narrative development of your results, thus choosing the most representative concerning each finding.
- Lines 295-296; 329-330: Please relocate these lines to the Discussion section since the Results section only inherits findings directly from your research.
DISCUSSION
- Throughout the discussion, there is mention of discrimination and negative perceptions experienced by your study population (LGBTQ+ older adults), somewhat caused by the formal care system. Essential elements in these experiences, such as the ageing process and healthcare services, are also highlighted. In this regard, ageism (discrimination based on age) may play a significant role, adding to gender discrimination as another element to discuss. I highly recommend that you read and consider using, if appropriate, the following study. It analyses discriminatory responses from the formal care system towards an older adult population. It discusses relevant elements to your research, such as the Stereotype Embodiment Theory (SET) and the Social Identity Theory (SIT):
- Martínez-Angulo, P., Muñoz-Mora, M., Rich-Ruiz, M., Ventura-Puertos, P. E., Cantón-Habas, V., & López-Quero, S. (2023). " With your age, what do you expect?": Ageism and healthcare of older adults in Spain. Geriatric Nursing, 51, 84-94.
- Consider introducing a brief paragraph as a “Conclusions” section and a “Strengths and Limitations” study section.
Author Response
Dear reviewer, Thank you very much for your guidance, it has been invaluable in my learning process. I have incorporated the suggestions you provided into my work, as shown in the PDF below. Additionally, I have incorporated your suggestions into this article. The parts in red font are modified based on your suggestions.
Reviewer 3 Report
Comments and Suggestions for Authors
Dear authors, thank you for allowing me to review this interesting manuscript focused on the long-term care service awareness, needs, and usage intentions of elderly male homosexuals in Taiwan.
Although I found this manuscript well-written and covering a timely topic, there are some points of concern that should be addressed before I could recommend it for publication. I try to summarize below some suggestions that could improve the quality of your manuscript.
Abstract, lines 29-30. You have suggested some "care techniques for homosexuals"; but what are these techniques? If there are no mentions about techniques (intented as practical skills) in the text I would suggest you to rephrase.
Introduction; page 2, lines 44-45. "Nevertheless, the government's focus on long-term care services for the LGBTQ+ community remains relatively limited." Is it a your assumption? Or is sustained by a reference? If it is an assumption please rephrase accordingly.
page 2, lines 84-86. "Governor Murphy stated that the government’s role is to protect these individuals from discrimination in long-term care settings because “No one should ever feel ashamed for who they are, and everyone should be able to live with the dignity and equality that they deserve." Is it relevant for your study? I would prefer a more deep description of what is the situation in Taiwan.
Methods; page 3, lines 122. This a major concern. Interviews are not a qualitative method but a data collection method. Please, reflect carefully about what qualitative design did you apply, because it is of outmost relevance for reporting your results and provide credibility to your study.
Table 1; I would suggest to add a column with the profession of each participant.
Research tools; it is not clear what questions you made, please include the interview guide you have used.
Data analysis; please provide a point-by-point description of the phases you have used to analyze data and provide methodological references that guided your process.
Results; without the presence of an interview guide and a clear methodological underpinning in my opinion these results are answers to your questions; my perception is that the results presented are mirroring the questions you have used, rather than a process of description/interpretation of their answers.
Please, in all results, check carefully if the quotations you provided are not overlapping with other themes, ore are out of their scope. An example is at page 5, lines 182-185 "I have experience with it since my family members and relatives have applied for home care or hired foreign caregivers. I understand that poor health can lead to many inconveniences; even bathing and eating require help. Therefore, I try my best to stay healthy and exercise as much as possible. I go to the gym to keep my body in good shape. (PE)" which is not connected with the information about long-term care services. But there are others, so take care of them.
Lastly, I would recommend not providing references in the results section (page 6, lines 252-253), there are not discussions.
Discussions. There are some parts that were already presented in the background section (e.g. page 9, lines 411-414).
Moreover, even if you have presented that participants were afraid of sexual abuses, you did not provide any discussion to this issue/potential issue.
I hope this series of suggestions will help you in making some amendments to your manuscript, thus improving its quality.
Regards
Author Response

(The authors gave the same response as above.)

Round 2
Reviewer 1 Report
Comments and Suggestions for Authors
No comments.
Author Response
Dear Reviewer,
I am truly grateful for their valuable feedback, which has contributed significantly to my growth.
Reviewer 2 Report
Comments and Suggestions for Authors
Congratulations on revising your manuscript and considering the suggestions made before. Nevertheless, there is still a need to address minor issues that I have attached now. I wish you the best of luck with your future and further research:
- The term “elderly” is still used occasionally throughout the manuscript. Please use “older people” or “older adults” instead of “elderly” to unify the terminology used throughout the study.
- Please consider using the expression “older adults male homosexuals”, as it is employed in the title already, instead of “elderly gay”, and unify this expression employed throughout the study.
- Lines 70-74 are duplicated, and the manuscript has some spelling mistakes. Please review all this throughout the study to fix this writing issue.
- Please in-cite the Tables made within the text.
Author Response
Dear reviewer, I am very grateful for their valuable feedback, which has contributed greatly to my growth. Based on your suggestion, I have replaced all "elderly" with "older adults" in the article. Following your suggestion, I have replaced all occurrences of "elderly gay" with "older adults gay men " in the manuscript. Based on your suggestions, I deleted the redundant parts and hired a professional English editor to edit the manuscript in English. As per your suggestion, I have added citations to the sources in the tables in the text. Thanks again for your guidance and encouragement. I will continue to work hard to improve! I wish you all the best and good health!
Reviewer 3 Report
Comments and Suggestions for Authors
Dear authors, thank you for allowing me to review this second version of this manuscript.
I appreciated you had considered all my comments and amended the manuscript accordingly. I found that the quality of presentation has improved significantly.
I would suggest the disclosure of the limits of the study at the end of the discussion section to give more trustworthiness to the reporting.
Regards
Author Response
Dear Reviewer,
I am truly grateful for their valuable feedback, which has contributed significantly to my growth.
I have added a discussion on the limitations of the study in the conclusion, as follows:
The study has lim-itations, including: 1. Sample Limitations: The participants may not fully represent the entire older adults gay men community in Taiwan, potentially limiting the generaliza-bility of the study results. 2.Social-Cultural Influences: The long-term care needs and experiences of Taiwan's older adults gay men are influenced by social-cultural factors, which may vary across different regions, ethnic groups, or socioeconomic strata, lim-iting the study's applicability. 3.Subjective Evaluation: The use of qualitative methods, such as in-depth interviews, may introduce subjective biases from researchers, poten-tially affecting the comprehensive reflection of participants' true circumstances.
Thank you once again for your guidance and encouragement. I will continue to strive for improvement! Wishing you all the best and good health!
